# Impact of Thrombopoietin Receptor Agonists on Pathophysiology of Pediatric Immune Thrombocytopenia

**DOI:** 10.3390/cimb47010065

**Published:** 2025-01-18

**Authors:** Paschalis Evangelidis, Konstantinos Tragiannidis, Eleni Gavriilaki, Athanasios Tragiannidis

**Affiliations:** 12nd Propedeutic Department of Internal Medicine, Hippocration Hospital, Aristotle University of Thessaloniki, 54642 Thessaloniki, Greece; pascevan@auth.gr (P.E.); elenicelli@yahoo.gr (E.G.); 2Children & Adolescent Hematology-Oncology Unit, Second Department of Pediatrics, School of Medicine, Aristotle University of Thessaloniki, 54124 Thessaloniki, Greece; konstantinos.tragiannidis@gmail.com

**Keywords:** eltrombopag, immune thrombocytopenia, pediatrics, romiplostim, thrombopoietin receptor agonists

## Abstract

Immune thrombocytopenia (ITP) in pediatric patients is a common cause of isolated thrombocytopenia. Various pathophysiological mechanisms are implicated in ITP pathogenesis, including the production of autoantibodies against components of platelets (PLTs) by B-cells, the activation of the complement system, phagocytosis by macrophages mediated by Fcγ receptors, the dysregulation of T cells, and reduced bone marrow megakaryopoiesis. ITP is commonly manifested with skin and mucosal bleeding, and it is a diagnosis of exclusion. In some ITP cases, the disease is self-limiting, and treatment is not required, but chronic-persistent disease can also be developed. In these cases, anti-CD20 monoclonal antibodies, such as rituximab and thrombopoietin (TPO) receptor agonists, can be used. TPO agonists have become standard of care today. It has been reported in the published literature that the efficacy of TPO-RAs can be up to 80% in the achievement of several end goals, such as PLT counts. In the current literature review, the data regarding the impact of TPO agonists in the pathogenesis of ITP and treatment outcomes of the patients are examined. In the era of precision medicine, targeted and individualized therapies are crucial to achieving better outcomes for pediatric patients with ITP, especially when chronic refractory disease is developed.

## 1. Introduction

Immune thrombocytopenia (ITP) is an acquired hematological disorder that is characterized by immune-mediated destruction and the impaired production of platelets (PLT), causing a notable reduction in PLTs count (less than 100,000/µL). This condition is characterized by hemorrhagic manifestations and, as a result, affects the patients’ quality of life (QoL) [1]. In the pediatric population, this condition is mostly self-limited, with improvement in PLT count after some time [2]. However, sometimes it can progress to chronic disease, and in these cases, long-term treatment is essential. The pathophysiological mechanisms for ITP in children constitute a field for further research. At the same time, recent studies have already confirmed the role of thrombopoietin (TPO) and megakaryocyte biology in the progression of the disease.

TPO is a key regulator for megakaryopoiesis and platelet production because its role is to bind to its receptor (c-Mpl) in order to stimulate megakaryocyte proliferation and platelet release from the bone marrow. Levels of TPO are not as increased as might be expected in ITP patients due to the high rates of platelet destruction, which highlights the autoimmune profile of the disease [3]. Also, the antibodies that target the PLTs often inhibit bone marrow megakaryocytes, leading to the suppression of PLT production. As a result, there is an important imbalance between the destruction and production of PLTs, which confirms the importance of TPO receptor agonists (TPO-RAs) usage as an option for ITP management in children [4]. 

TPO-RAs, such as eltrombopag and romiplostim, are synthetic agents that stimulate megakaryocyte activity and increase platelet production [5]. They are used in both pediatric and adult populations, and they have confirmed their efficacy in increasing PLT counts. In addition, they appear to have further important effects in managing ITP, such as in the bone marrow microenvironment, by promoting megakaryocyte maturation and reducing the immune-mediated inhibition of megakaryopoiesis [6]. Overall effects are of paramount importance, especially in pediatric patients.

However, a challenging field for research is the impact that TPO-RAs have on the immune regulation of children since chronic ITP is often associated with a breakdown of immune tolerance, which results in the production of autoantibodies against platelet antigens. Some existing studies show that TPO-RAs may help restore immune homeostasis, but there is no further information concerning pediatric patients [7,8]. A narrative review of the literature was performed using PubMed and Medline search engines to identify original articles. In our search, full-text articles published in English were included, and all the available data were examined critically. The keywords of research were “immune thrombocytopenia” or “ITP” combined with “pediatric”, “TPO-RA”, “eltrombopag”, or “romiplostim”.

## 2. Pediatric ITP: Clinical Approach

### 2.1. Clinical Manifestations and Epidemiology of Pediatric ITP

ITP in pediatric patients can be divided into the following categories: “newly diagnosed ITP” refers to cases in which thrombocytopenia is present for 3 months from the diagnosis, “persistent ITP” is defined as thrombocytopenia lasting 3 to 12 months, and “chronic ITP” describes thrombocytopenia lasting for more than 12 months [9,10]. The term “severe ITP” is used for patients with clinically significant bleeding manifestations at diagnosis that require treatment and for those with new-onset bleeding tendencies after diagnosis [9].

In the majority of pediatric patients with ITP, children are asymptomatic or exhibit bruising, petechiae, or epistaxis [11]. Bleeding in oral mucosal, menorrhagia in adolescents, macroscopic hematuria, gastrointestinal (GI) bleeding, and rarely intracranial bleeding [12]. The incidence of intracranial hemorrhage is below 1% and is associated with high mortality and morbidity [13]. Bleeding manifestations are more prevalent in adolescent patients, while asymptomatic disease is mainly observed between 1 and 3 years old [12]. Fatigue is also an important issue in children with ITP, contributing to lower quality of life levels [14].

Often in ITP pediatric cases, the disease is self-limited and benign, and in 75% of patients it resolves within a 6 month period [11]. Kühne and colleagues in their observational study included 1496 children with ITP and found that the highest incidence of ITP is observed during spring and early summer [15]. ITP incidence in children has been described between 1.9 and 8.8 cases per 100,000 [13,16]. Moreover, the median age of diagnosis might be lower in males in comparison to females [13].

### 2.2. Diagnosis and Secondary ITP Causes

ITP is a diagnosis of exclusion, and detailed personal and family history for bleeding and thorough clinical examination is essential [15,17]. In every case of thrombocytopenia, pseudothrombocytopenia should be excluded through a blood smear examination [18]. After the establishment of a thrombocytopenia diagnosis, a differential diagnosis of the causes of ITP should be performed. In over half of ITP cases, a recent infection is reported in the patient’s history [19]. At the same time, associations between the measles–mumps–rubella (MMR) vaccine and ITP development have been described [20]. Rarely, COVID-19 infection and vaccination against COVID-19 can cause a syndrome similar to ITP, accompanied by the development of thrombotic events in some cases, known as vaccine-induced thrombotic thrombocytopenia [21,22,23,24]. It is important that ITP in children is differentially diagnosed by congenital causes of thrombocytopenia, such as Wiskott–Aldrich syndrome, congenital amegakaryocytic thrombocytopenia, and thrombocytopenia absent radia syndrome [25]. These syndromes are often connected with skeletal abnormalities and other clinical manifestations [26].

In adults with ITP, hepatitis B and C virus, human immunodeficiency virus, and Helicobacter pylori infections should be ruled out, while testing for these pathogens in pediatric patients is not essential and should be performed based on epidemiological and clinical data [27,28]. Testing for common variable immunodeficiency with measurement of quantitative immunoglobulins (Ig IgG, IgA, IgM) is recommended for children with ITP [28,29]. In some cases, ITP might be the result of underlying autoimmune disorders, such as antiphospholipid syndrome or systemic lupus erythematosus (SLE) [30]. Higher titers of antinuclear antibodies (ANAs) at the timepoint of ITP diagnosis have been associated with SLE development [31]. In Table 1, the underlying causes of ITP in children are presented. In Evan’s syndrome, thrombocytopenia is combined with one or more cytopenias, which commonly include autoimmune hemolytic anemia. In Evan’s syndrome, direct antiglobulin tests are positive [32]. Antiplatelet antibodies (glycoprotein GPIIb-IIIa autoantibodies) have a high specificity for ITP, but their sensitivity is low for the diagnosis of ITP [33]. Thus, testing for these antibodies is not routinely recommended [17]. Furthermore, bone marrow aspirate and biopsy are not suggested in children with ITP, unless the exhibit abnormal findings in clinical or blood smear examinations [17]. Bone marrow examinations might also be indicated for patients with refractory disease.

### 2.3. Refractory ITP

In the majority of pediatric ITP cases, even in cases with severe thrombocytopenia, PLT counts are normalized within 12 months of diagnosis [34]. Moreover, in most children with ITP, spontaneous remission is common, and response rates to first (IVIG and corticosteroids)- and second (rituximab, TPO-receptor agonists)-line treatments are high [35,36,37]. However, despite this, some will have a refractory disease. In the systematic review of Ibrahim et al., 11 studies proposing different definitions for refractory pediatric ITP were identified, without an agreement among them [38]. Inadequate treatment response and specified PLT count thresholds were used in most of the published studies to define refractory ITP [38]. Given the various definitions used in the different studies, it is difficult to estimate the real incidence of refractory ITP. Recently, the Intercontinental ITP Study Group and the Pediatric ITP Consortium of North America (ICON) proposed a novel definition for refractory ITP [39]. In Table 2, the definition of refractory ITP by the Intercontinental ITP Study Group and the Pediatric ICON is presented. The authors also highlight that some patients might not belong to either of these categories. In these patients those who need a longer duration of first-line treatments to respond and those who relapse while on second-line therapies are included. It is considered crucial to evaluate the incidence of refractory ITP in real-world settings using this novel definition.

## 3. Immune Dysregulation in ITP

In the pathophysiology of ITP, both increased destruction-reduced lifespan and decreased production of PLTs are implicated [40,41,42,43]. Various mechanisms have been proposed as pathogenic in this clinical entity, including the production of antibodies against PLT components, the apoptosis of PLTs, and the activation of T cell-mediated immunity [44]. 

Specifically, circulating autoantibodies against glycoprotein Ib and IIb/IIIa, secreted by B-cells, bind to antigens on the surface of PLTs. These antibody-coated PLTs are destructed by activated macrophages in the spleen and other reticuloendothelial tissues, such as the liver [45,46]. The activation of Fcγ receptors through the spleen tyrosine kinase results in the phagocytosis of PLTs [47,48]. Moreover, the classical pathway activation of the complement system leads to the destruction of the antibody-coated thrombocytes [49]. Autoantibodies have also been proposed as suppressors of PLT production by bone marrow megakaryocytes [33,40,47]. Modifications in the glycans, and mainly loss of terminal sialic acid, might also be implicated in ITP pathogenesis. It has been suggested that antibodies against glycoprotein Ib lead to loss of sialic acid (desialylation), leading to their uptake by the Ashwell–Morell receptor, the activation of the JAK2-STAT3 signaling pathway, the increased production of TPO, and the induction of PLT production in the bone marrow [50,51,52,53].

Macrophages present PLT antigens of the major histocompatibility complex class II to the T cell receptors of autoreactive T cells [54]. Thus, the T cell regulatory response is reduced, while the activity of type 1 T-helper (Th1) and 17 T-helper (Th17) types and cytotoxic T cells is enhanced, leading to the destruction of PLTs and megakaryocytes [55,56,57]. Zufferey et al., in their study, showed that mature megakaryocytes can present antigens to CD8+ T (cytotoxic) cells, mediating ITP in vivo [58]. PLT-derived extracellular vehicles have been shown to promote the differentiation and activation of CD8+ T cells by inserting antigens with major histocompatibility [59]. Moreover, in ITP patients, increased PLT apoptosis has been described and might be the result of the dysregulation of Bcl-xL (an antiapoptotic protein) and Bax expression [50,60,61]. 

Autoantibodies might play a crucial role in the pathogenesis of ITP, but in up to 40% of ITP patients, antibodies against PLTs are not detectable, as shown in the metanalysis of Vrbensky et al. [62]. In such cases, defects in other components of the immune system might be implicated, such as in cytokines, chemokines, the complement system, and antigen-presenting, natural killer, and T cells [63,64,65,66,67,68]. In particular, the dysfunction of T cells might result in the desialylation of the PLTs [65]. In Figure 1, the basic process of ITP pathogenesis and the role of targeted therapeutics are summarized.

## 4. Treatment of ITP in Pediatrics

Treatment of pediatric ITP can be challenging since there are no clear guidelines and therapy varies, depending on the patient’s clinical status condition. In most cases, PLT counts in children normalize after some time. In cases where the condition is not improved, treatment is needed. ITP treatment options are categorized into front-line therapies (corticosteroids and IVIg) and second-line therapies (Rituximab and TPO-RA), which are preferred to treat chronic or refractory ITP [38]. In very severe and rare cases only, ITP can be life-threatening. Negative effects on the ITP patient’s QoL, especially in childhood, have been described [69].

### 4.1. Front-Line Treatment: Corticosteroids and IVIg

Corticosteroids are commonly used for the treatment of ITP. These, as a first-line treatment, constitute a therapeutic option for the initial therapeutic period, since most cases of ITP are considered self-limited. Frequently used corticosteroids are prednisone, prednisolone, or dexamethasone [70,71]. The efficacy of corticosteroids (improvement in PLT counts) is confirmed in 70–80% of pediatric cases [10]. This highlights that sometimes corticosteroids can also be part of combined therapy regimens in order to achieve better therapeutic outcomes. Their aim is to suppress the immune response and reduce autoantibody production. Corticosteroids decrease the phagocytosis of antibody-coated platelets through macrophages in the spleen and inhibit the production of platelet-targeting autoantibodies [72]. Prednisone, the most common corticosteroid used, requires a dose of 1–2 mg/kg daily for 2 weeks, while dexamethasone (0.6 mg/kg/day) is given for four consecutive days [73]. However, long-term administration is not suggested due to the side effects that can present.

In addition to corticosteroids, IVIg can also be an option as a front-line therapy, especially in children with more severe thrombocytopenia. IVIg reduces PLT destruction by saturating the Fc receptors on splenic macrophages [74,75]. It is administered as a single dose (0.8–1 g/kg) or divided into 2 days (0.4/g/kg/day), with PLT counts improving immediately after administration [73]. IVIg acts faster and is preferred in emergency cases, but side effects, such as headache, nausea, or even hemolysis after the infusion, have to be considered [10]. 

The choice between corticosteroids and IVIg as front-line therapy depends on several factors, like the urgency of PLT improvement, the clinical manifestation of the patient, and the severity of the ITP. In both treatments, side effects can be observed, and both have a risk of failure, especially when using steroids [10]. In their study, Cao and colleagues used plasma proteomics to predict the prednisolone treatment outcomes in pediatric ITP and found that myosin heavy chain 9 and fetuin B levels were significantly lower in prednisolone-resistant patients [76]. Also, another option for emergency cases with uncontrolled bleeding is to choose a combination treatment with prednisone and IVIg [77]. 

Regarding the response to IVIG, it has been shown that patients who are homozygous for the FcGR2B-232I allele (encoding Fc gamma receptor) are more likely to respond compared to those homozygous for the FCGR2B-232T [35]. In the study by Peng et al., the presence of anti-glycoprotein Ib/IX antibodies has been associated with lower response rates to IVIG treatment [78]. Schmidt and colleagues have developed a risk score for the prediction of response to IVIG treatment, incorporating five variables: hemoglobin, PLT count, the identification of anti-PLT antibodies, genetic polymorphisms Fc-receptor IIc, and a history of recent vaccination [79]. Moreover, the Childhood ITP Recovery Score Calculator, incorporating only clinical variables (age, PLT count at diagnosis, sex, history of infection, history of vaccination, disease onset, and clinical manifestations), has been validated as a predictor of transient and persistent ITP in children [80]. 

Respondents to treatment are considered those who exhibit a doubling of their PLT count and PLTs between 30 × 10^9^/L and 100 × 10^9^/L, while a complete response is attained when PLT is over 100 × 10^9^/L. No response is considered when PLT counts are less than 30 × 10^9^/L or half of the PLT values before the treatment initiation. Resolving bleeding symptoms is also important for the evaluation of response [81]. 

### 4.2. The Role of Anti-CD20 Monoclonal Antibodies

In cases where ITP progresses to a chronic condition, long-term treatment is required. In such cases, rituximab, a monoclonal CD20 antibody, which causes peripheral B-cell depletion, is among the most common first-line treatment choices. It has been used in patients with lymphoma as an anti-neoplastic agent for the past few decades [69]. Rituximab achieves an initial response rate (platelet count ≥ 50 × 10^9^/L) of 50–60% and a 5-year sustained response of 25–30% [82]. Its efficacy is underlined in several studies and factors associated with better treatment outcomes include female gender, younger age, and prior response to corticosteroids [83,84]. It is generally considered a safe agent, even though there are some risks that should be evaluated, especially in patients with immunodeficiency or hypogammaglobulinemia and neutropenia [85]. Rituximab reduces anti-platelet antibody production, and it is used for various autoimmune disorders beyond ITP [69]. Harris et al. assessed the effectiveness and safety of rituximab in the pediatric and adult population and concluded that 58% of patients of < 18 years old met the criteria for complete response [86]. Regarding the effects after the infusion, as seen in several studies with a significant number of patients, it can be concluded that almost all the adverse effects were mild and only a few could be characterized as severe [87,88,89]. The systematic review of Yi Liang et al., which summarized the published data regarding rituximab use in pediatrics, concluded that the pooled complete response rate of rituximab was 39% [90]. The response rate of rituximab was also evaluated in more recent studies [37,91,92].

### 4.3. TPO-RAs in Pediatric ITP Management 

TPO-RAs are also included in the second-line treatment options in pediatric ITP [93]. The main TPO agonists used in children (over 1 year old) are eltrombopag and romiplostim [94]. Investigations into the safety and efficacy of avatrombopag have started, but it is not yet approved for the pediatric population. For both eltrombopag and romiplostim, it is necessary to monitor for side effect development. At the same time, their efficacy and safety have been described in various studies [92,95,96,97].

Eltrombopag is administered orally and is approved for aplastic anemia management as well as ITP. It binds to the transmembrane region of the TPO receptor, stimulating platelet production [98]. The dosage for Eltrombopag in children is 25 mg/day if the child’s weight is less than 27 kg or 50 mg daily if the child’s weight is over 27 kg. Eltrombopag’s safety has been demonstrated in various studies [95,99,100]. Mainly, monitoring for hepatotoxicity and bone marrow toxicity is required, while other effects are mild, even though in adult patients more severe adverse events, such as cataracts, have been described [8,101]. 

Romiplostim is administered as a subcutaneous injection. For this reason, its use in pediatric patients might be difficult. It acts as a peptibody that mimics TPO and binds to the extracellular domain of the TPO receptor [101,102]. It is administered once a week, and the initial dose is 1 μg/kg/week [103]. Adverse events of romiplostim administration are also mild, but rare cases of bone marrow fibrosis have been described. Its safety and efficacy are similar to eltrombopag, as confirmed in several studies [95,99,100]. In the systematic review of Oliveira et al., 2023, in which two randomized controlled trials were included, it was shown that romiplostim can improve durable and overall PLT response in children with ITP, compared to the placebo [104]. 

For the discontinuation of TPO-RAs treatment, Marcos-Peña et al. suggested that the general goal is to reach a PLT count of ≥ 80–100 × 10^9^/L for 3 months, or after 9 months with a response, and the shared-approval of the patient also is essential [100]. A period of close monitoring is crucial, post treatment discontinuation, to avoid PLT dropping below 20 × 10^9^/L [100]. Eltrombopag and romiplostim present similar efficacy and response rates, while some minor differences in the mechanism of action and effects have been recognized. The choice between them is based on the preferred route of administration along with the comorbidities of the pediatric patient. Further and larger studies should be performed on TPO-RAs’ long-term efficacy in children with ITP. Moreover, future research should focus on the development of predictive models based both on clinical and laboratory variables for a response to TPO-RAs. Multicenter collaboration is essential in this field. In Table 3, randomized controlled trials examining the efficacy and safety of eltrombopag and romiplostim are presented. Moreover, in Figure 2, an algorithm for the management of pediatric ITP is presented. 

## 5. Novel Therapeutics in ITP 

Signaling via the Fc-gamma receptor and Syk kinase, as was mentioned above, is of paramount importance in ITP pathogenesis [108]. Fostamatinib is an oral inhibitor of the Syk molecule, approved for the treatment of ITP in adult patients by the Food and Drug Administration (FDA) [109]. In the phase 3 trial of Bussel et al., 146 patients were treated with fostamatinib and 44% of them achieved an overall response (defined as ≥1 platelet count ≥ 50,000/μL between weeks 1–12 of the treatment) [110]. The efficacy and safety of fostamatinib were confirmed by the open-label extension of this trial [111]. Diarrhea and hypertension are the main adverse events reported in patients who receive this agent. Use of fostamatinib is contradicted in children and adolescents with ITP and is limited only to adulty populations.

Bruton’s tyrosine kinase (BTK) is also expressed by PLTs, and rilzabrutinib, a BTK inhibitor, which reduces macrophages-induced PLT destruction, has been investigated for the management of ITP. In a phase 1–2 trial of 60 adult ITP patients, rilzabrutinib was found to be safe and effective, with a response rate of approximately 40% [112,113]. Recently, in the 2024 meeting of the American Society of Hematology (ASH), these results were confirmed in a phase 3 study by Kuter et al. (NCT04562766) in adults and adolescents with ITP [114]. The most prevalent adverse events included diarrhea, nausea, and headaches. The efficacy and safety of this agent in children with refractory ITP should be investigated in future studies. 

Avatrombopag, an oral agent acting as a transmembrane TPO receptor agonist, has shown similar efficacy to eltrombopag, and its use has been approved by the FDA for adults with ITP [115]. There are emerging data that avatrombopag is a safe and effective agent for the management of chronic and persistent ITP in pediatrics [116,117]. In Table 4, the data regarding the use of avatrombopag in pediatric patients are presented. Recently, daratumumab, an anti-CD38 monoclonal antibody that is used for the treatment of multiple myeloma, combined with avatrombopag, was used safely and effectively in a child with chronic and persistent ITP [118]. Moreover, herombopag, a second-generation TPO agonist, which has been shown to be effective in post-allogeneic hematopoietic cell transplantation thrombocytopenia management, is being investigated in children with chronic and persistent ITP (NCT05685420) [119,120]. Obinutuzumab is the first personalized type II glycosylation-engineered CD20 monoclonal antibody, and its safety and efficacy for pediatric ITP is under examination (NCT06094881) [117,121]. More data regarding the cost-effectiveness and long-term safety of these novel agents in pediatric settings are essential.

## 6. Conclusions

In this review, an up-to-date approach to the impact of TPO agonists in the treatment and pathophysiology of pediatric ITP was used. In many cases, ITP in children is self-limiting and treatment is not required, while in other cases, a persistent and chronic disease could develop. For these cases, beyond anti-CD20 monoclonal antibodies, TPO agonists have become the standard of care. Improvements have been described, not only in PLT counts but also in the quality of life of these vulnerable patients. The different outcomes observed in the different studies could be attributed to regional variations. Multicenter studies evaluating the efficacy of these agents could be helpful in order to obtain safe conclusions. Next-generation therapeutics, targeting Fcγ receptors and second-generation TPO agonists are under investigation for pediatric ITP management. Future studies should focus on the development models that will be helpful in the prediction of outcomes of patients who are treated with TPO agonists. 

## Figures and Tables

**Figure 1 cimb-47-00065-f001:**
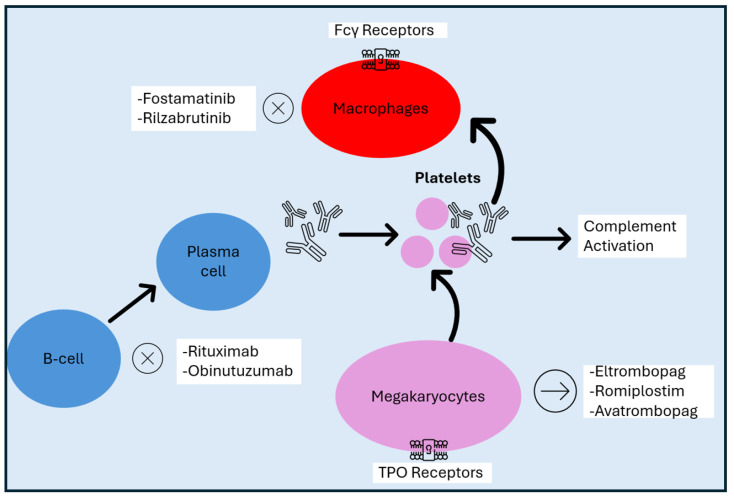
ITP pathogenesis and the role of novel therapeutics. TPO: thrombopoietin; ITP: immune thrombocytopenia.

**Figure 2 cimb-47-00065-f002:**
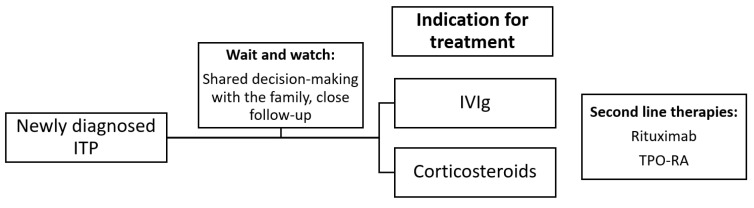
A practical algorithm for pediatric ITP management. ITP: immune thrombocytopenia; IVIg: intravenous immunoglobulin; TPO-RA: thrombopoietin receptor agonist.

**Table 1 cimb-47-00065-t001:** Underlying causes of ITP in pediatric patients.

Cause	Disease Examples
Infections	Viral infections (EBV, influenza, VZV, CMV, HIV, COVID-19)
Vaccinations	MMR and, rarely, vaccinations against varicella, hepatitis A, pneumococcus, tetanus–diphtheria–acellular pertussis vaccines
Autoimmune disorders	SLE, APS
Primary immunodeficiency syndromes	Common variable immunodeficiency, DiGeorge (22q11.2 deletion) syndrome
Drugs	
Lymphoproliferative disorders and other malignancies	Leukemia, Myelodysplastic syndrome, autoimmune lymphoproliferative syndrome, non-Hodgkin/Hodgkin Lymphoma

EBV: Epstein–Barr virus, VZV: varicella zoster virus, CMV: cytomegalovirus, HIV: human immunodeficiency virus, MMR: measles, mumps, and rubella, SLE: systemic lupus erythematosus, APS: antiphospholipid syndrome.

**Table 2 cimb-47-00065-t002:** Definition of refractory ITP by the Intercontinental ITP Study Group and the Pediatric ITP Consortium of North America [39].

Refractory ITP	Criteria
Newly diagnosed refractory	Pediatric patients who Still require treatment; Had no response to at least two first-line agents at standard dosing: steroids (>1 mg/kg/dose) for 4 days or more and IVIG (0.8–1 g/kg); PLTs count < 20 × 10^9^/L, 1 week post-treatment.
Persistent/chronic refractory	Pediatric patients who do not respond to at least two second-line treatments of different categories (such as rituximab and/or TPO-RA), independently from the response to first-line treatment.

ITP: immune thrombocytopenia, IVIG: intravenous immunoglobulin, PLTs: platelets, TPO-RA: thrombopoietin-receptor agonists.

**Table 3 cimb-47-00065-t003:** Randomized controlled trials examining the efficacy and safety of eltrombopag and romiplostim.

First Author, Year of Publication, Reference	Agent	Number of Participants	Outcomes	Adverse Events	Limitations
Bussel, 2015, [105]	Eltrombopag	45 patients received eltrombopag	From weeks 1 to 6, 28 (62%) patients who received eltrombopag achieved the primary endpoint of platelet count 50 × 10^9^ per L in comparison to 32% in the placebo group (*p* = 0.011).	Headache, upper respiratory tract infections, and diarrhea	Conservative approach to the initial dosing
Grainger, 2015, [106]	Eltrombopag	63 patients received eltrombopag	A total of 25 (40%) patients who received eltrombopag achieved the primary outcome of platelet counts of at least 50 × 10^9^ per L for 6 of the last 8 weeks of the trial compared with 3% patient in the placebo group (*p* = 0.0004).	Nasopharyngitis, rhinitis, upper respiratory tract infections, and cough	Use of the WHO bleeding scale
Bussel, 2011, [107]	Romiplostim	17 patients received romiplostim	A total of 15 of the 17 (88%) patients in the romiplostim group achieved the efficacy endpoints of a platelet count of 50 × 10^9^/L or greater for two consecutive weeks and an increase in platelet count of 20 × 10^9^/L or greater above baseline for two consecutive weeks. A significantly higher number of patients in the romiplostim achieved the two endpoints in comparison to the placebo group (*p* = 0.0008 for each endpoint).	Headache, epistaxis, oropharyngeal pain, pyrexia, contusion, rash, cough, and vomiting	Small sample size, minor differences between control and romiplostim group
Tarantino, 2016, [103]	Romiplostim	42 patients received romiplostim	A more durable platelet response was seen in 22 (52%) patients in the romiplostim group than in comparison with 10% in the placebo group (*p* = 0.002).	Headache and thrombocytosis	Disparity between the incidence of serious adverse events between the placebo and control group

**Table 4 cimb-47-00065-t004:** Data regarding the use of avatrombopag in pediatrics.

First Author, Year of Publication, Reference	Study Design	Number of Participants	Patients Characteristics	Outcomes	Adverse Events	Limitations
An, 2023 [116]	Retrospective study	20	8 male, 12 females, median age 7.3 years	At day 90 of therapy, a platelet response (≥50 × 10^9^/L) was observed in 93% of the patients (*p* < 0.01 compared to control group).	One case of headache, two cases of epistaxis and petechia	Not reported
Cheng, 2023 [122]	Retrospective studies	11	7 males, 4 females, median age 8.3 years	Of the patients, 81.8% (9/11) and 54.6% (6/11) experienced an overall and complete response, respectively. The median PLT count was significantly increased from eltrombopag to avatrombopag (*p* = 0.007).	Vomit, diarrhea, headache, nasopharyngitis	Retrospective design, small study sample
Turudic, 2024 [123]	Case report	1	2-year-old, ANA-positive ITP	A complete response was achieved on day 74 (>100 × 10^9^/L).	-	Case report
Wang, 2024 [117]	Retrospective study	34	18 males, 16 females, mean age 6.3 years	An overall response was achieved in 79.4% patients and a complete response in 67.7%.	Upper respiratory infections, fever, gastrointestinal symptoms	Retrospective observational study, absence of control group, small sample size

ANA: antinuclear antibodies; ITP: immune thrombocytopenia; PLT: platelet.

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
