# Peer review of "Impact of Thrombopoietin Receptor Agonists on Pathophysiology of Pediatric Immune Thrombocytopenia"

_cimb, 2025, doi:10.3390/cimb47010065_

Round 1

Reviewer 1 Report

Comments and Suggestions for Authors

I consider that the main advantage is that this compilation of information helps to update not only specialists such as hematologists and pediatric hematologists but also other specialists who eventually treat this pathology.

Author Response

I consider that the main advantage is that this compilation of information helps to update not only specialists such as hematologists and pediatric hematologists but also other specialists who eventually treat this pathology.

Response: We would like to thank the reviewer for the time dedicated to reviewing our work. Moreover, we are grateful for the positive feedback.

Reviewer 2 Report

Comments and Suggestions for Authors

This is an interesting review addressing the impact of TPO agonists in the treatment and pathophysiology of pediatric ITP.

This review is complete, with up to date references. 

What could be added are the following references:

- Ther Adv Hematol. 2020 Apr 28;11:2040620720912992. doi: 10.1177/2040620720912992;

- Blood. 2022 Aug 11;140(6):542-555. doi: 10.1182/blood.2020006480;

- LETTER TO THE EDITOR European J of Haematology - 2024 - DOI: 10.1111/ejh.14262.

Lines 183 and 285: the term “agonists” could be removed.

Author Response

This is an interesting review addressing the impact of TPO agonists in the treatment and pathophysiology of pediatric ITP.  

This review is complete, with up to date references.

Response: We would like to thank the reviewer for the positive comments.

What could be added are the following references:

- Ther Adv Hematol. 2020 Apr 28;11:2040620720912992. doi: 10.1177/2040620720912992;

- Blood. 2022 Aug 11;140(6):542-555. doi: 10.1182/blood.2020006480;

- LETTER TO THE EDITOR European J of Haematology - 2024 - DOI: 10.1111/ejh.14262.

Response: Thanks for this idea! We have included these references in the revised version of the manuscript.

Lines 183 and 285: the term “agonists” could be removed.

Response: Thanks for this comment! We have removed these terms in the revised version of the manuscript.

Reviewer 3 Report

Comments and Suggestions for Authors

The manuscript presented for review titled “Impact of Thrombopoietin Receptor Agonists on Pathophysiology of Pediatric Immune Thrombocytopenia" provides a comprehensive review of the role of thrombopoietin receptor agonists (TPO-RAs) in the treatment and pathophysiology of pediatric immune thrombocytopenia (ITP). This study successfully highlighted the clinical and pathophysiological aspects of ITP and the importance of individualized therapy.

The subject matter of this article is important as it emphasizes the importance of precision medicine in tailoring therapies for pediatric patients, aligning with current trends in patient-centered care.

The systematic analysis of the topics raised by the Authors has been presented in a clear and coherent manner. The language of the work is understandable and easy to read. The manuscript is generally well written and clear.

However, certain areas of the manuscript require further clarification to enhance its overall quality.

Abstract

Authors should incorporate in abstract specific data points from clinical trials to demonstrate the clinical relevance of TPO-Ras (e.g. percentage of patients achieving platelet counts above a target threshold or reductions in bleeding episodes)

Methodology

The manuscript effectively summarizes current studies but would benefit from a deeper exploration of their limitations, such as small sample sizes, potential biases in study design, or variations across regions.

Also, in Table 3 and 4 including columns for study limitations or statistical significance would be beneficial.

While the paper emphasizes pediatric ITP, several sections include findings from adult studies without clearly identifying their relevance. For instance, "In a phase 1-2 trial in 60 adult ITP patients, rilzabrutinib was found safe and effective, with a response rate of approximately 40%."; ""Fostamatinib is an oral inhibitor of the Syk molecule, approved for the treatment of ITP in adult patients by the Food and Drug Administration (FDA)... The use of Fostamatinib is contradicted in children and adolescents with ITP and is limited only to adult populations.".... Authors should consider clearly differentiating the relevance of adult data or limiting the discussion only to pediatric studies.

Also, are there regional variations in incidence or treatment outcomes that could be highlighted?

Novel therapeutics in ITP

The manuscript mentions emerging therapies but lacks depth in evaluating pediatric-specific applications. Authors should explore more cost-effectiveness, accessibility, and long-term safety.

Authors should consider adding a section on how clinicians can apply these findings in practice which would enhance the manuscript’s utility. Highlighting approaches to patient selection and presenting clear treatment algorithms would offer practical guidance, ensuring the research is more directly applicable to clinical decision-making.

Author Response

The manuscript presented for review titled “Impact of Thrombopoietin Receptor Agonists on Pathophysiology of Pediatric Immune Thrombocytopenia" provides a comprehensive review of the role of thrombopoietin receptor agonists (TPO-RAs) in the treatment and pathophysiology of pediatric immune thrombocytopenia (ITP). This study successfully highlighted the clinical and pathophysiological aspects of ITP and the importance of individualized therapy.

The subject matter of this article is important as it emphasizes the importance of precision medicine in tailoring therapies for pediatric patients, aligning with current trends in patient-centered care.

The systematic analysis of the topics raised by the Authors has been presented in a clear and coherent manner. The language of the work is understandable and easy to read. The manuscript is generally well written and clear.

Response: We are deeply thankful for the reviewer’s comments and feedback. Your comments and ideas were valuable for the quality of our manuscript.

However, certain areas of the manuscript require further clarification to enhance its overall quality.

Abstract

Authors should incorporate in abstract specific data points from clinical trials to demonstrate the clinical relevance of TPO-Ras (e.g. percentage of patients achieving platelet counts above a target threshold or reductions in bleeding episodes)

Response: Thanks for this idea! We have included the following phrase in the revised version of the abstract.

“It has been reported in the published studies that the efficacy of TPO-RAs is up-to 80% in the achievement of PLT counts.”

Methodology

The manuscript effectively summarizes current studies but would benefit from a deeper exploration of their limitations, such as small sample sizes, potential biases in study design, or variations across regions.

Also, in Table 3 and 4 including columns for study limitations or statistical significance would be beneficial.

Response: We would like to thank the reviewer for this very interesting idea! We included in tables 3 and 4 one extra column regarding the limitations of each study. Moreover, we described the statistical significance of each of them.  

While the paper emphasizes pediatric ITP, several sections include findings from adult studies without clearly identifying their relevance. For instance, "In a phase 1-2 trial in 60 adult ITP patients, rilzabrutinib was found safe and effective, with a response rate of approximately 40%."; ""Fostamatinib is an oral inhibitor of the Syk molecule, approved for the treatment of ITP in adult patients by the Food and Drug Administration (FDA)... The use of Fostamatinib is contradicted in children and adolescents with ITP and is limited only to adult populations.".... Authors should consider clearly differentiating the relevance of adult data or limiting the discussion only to pediatric studies.

Response: The reviewer is right. The following phrase was included in the revised manuscript:

“The efficacy and safety of this agent in children with refractory ITP should be investigated in future studies.”

Also, are there regional variations in incidence or treatment outcomes that could be highlighted?

Response: Indeed the reviewer is right. The following phrase was added in the novel manuscript:

“The different outcomes observed in the different studies might be attributed to regional variations. Multicenter studies evaluating the efficacy of these agents can be helpful in order to make safe conclusions.”

Novel therapeutics in ITP

The manuscript mentions emerging therapies but lacks depth in evaluating pediatric-specific applications. Authors should explore more cost-effectiveness, accessibility, and long-term safety.

Response: The reviewer is right. The following data were incorporated in the revised manuscript:

“More data regarding the cost-effectiveness and long-term safety of these novel agents in pediatric settings are essential.”

Authors should consider adding a section on how clinicians can apply these findings in practice which would enhance the manuscript’s utility. Highlighting approaches to patient selection and presenting clear treatment algorithms would offer practical guidance, ensuring the research is more directly applicable to clinical decision-making.

Response: Thanks for this interesting idea. We have included a figure, as you suggested (Figure 2). Thanks once again for your positive feedback and comments.

Reviewer 4 Report

Comments and Suggestions for Authors

The present review describes the data regarding the impact of TPO agonists in the pathogenesis of ITP and treatment of the patients.

The manuscript is well written, has many apropriate references. It maybe useful for scientists looking for the latest information.

However, it needs to include the methodology of data searching-how many publications were considered, where they were found, why cited references has been chosen? etc. Such methodology will help for increase interest in this review.

In summary, the manuscript needs the description of methodology.

Author Response

The present review describes the data regarding the impact of TPO agonists in the pathogenesis of ITP and treatment of the patients.

The manuscript is well written, has many apropriate references. It maybe useful for scientists looking for the latest information.

Response: We would like to thank the reviewer for the positive feedback. The reviewer’s comments were substantial for our work.

However, it needs to include the methodology of data searching-how many publications were considered, where they were found, why cited references has been chosen? etc. Such methodology will help for increase interest in this review.

In summary, the manuscript needs the description of methodology.

Response: We are thankful for this comment. The following section was added in the manuscript.

“A narrative review of the literature was performed using PubMed and Medline search engines to identify original articles. Full-text articles published in English, based on our search, were included, and all the available data were examined critically. The keywords of research were “immune thrombocytopenia” or “ITP” combined with “pediatric”, “TPO-RA”, “eltrombopag”, or “romiplostim”.”

Round 2

Reviewer 3 Report

Comments and Suggestions for Authors

I am pleased with how the authors addressed the comments and suggestions provided during the initial review of the manuscript titled " Impact of Thrombopoietin Receptor Agonists on Pathophysiology of Pediatric Immune Thrombocytopenia " . The authors have effectively addressed the feedback, leading to significant improvements in the manuscript's clarity, organization, and depth. Their responses were well-considered, effectively addressing all major concerns.